# Application of Teeth in Toxicological Analysis of Decomposed Cadavers Using a Carbamazepine-Administered Rat Model

**DOI:** 10.3390/diagnostics13020311

**Published:** 2023-01-14

**Authors:** Hiroaki Ichioka, Urara Saito, Kaori Shintani-Ishida, Takahira Shirahase, Nozomi Idota, Narisato Kanamura, Hiroshi Ikegaya

**Affiliations:** 1Department of Forensic Medicine, Graduate School of Medical Science, Kyoto Prefectural University of Medicine, 465, Kajii-cho, Kamigyo-ku, Kyoto 602-8566, Japan; 2Department of Dental Medicine, Graduate School of Medical Science, Kyoto Prefectural University of Medicine, 465, Kajii-cho, Kamigyo-ku, Kyoto 602-8566, Japan

**Keywords:** autopsy, blood concentration, carbamazepine, forensic odontology, fresh frozen sections, MS imaging, postmortem biochemical analysis, teeth, toxicant, toxicology

## Abstract

In a regular autopsy, blood and organs are used to quantify drug and toxicant concentrations; however, specimens such as blood cannot be collected from highly decomposed corpses, making the quantification of drug and toxicants impossible. This study aimed to estimate the blood carbamazepine (CBZ) concentration from teeth, a part of the human body that is best preserved after death. We sampled teeth and blood of rats administered CBZ. The correlation between the tooth and serum CBZ concentrations was analyzed. Rats were euthanized after CBZ administration and kept at 22 °C for 0 to 15 days before sampling the teeth and measuring the CBZ concentration. Undecalcified, fresh, frozen sections of rat teeth were prepared, and CBZ localization was evaluated. CBZ concentrations in both teeth and cardiac blood peaked at 60 min after administration and increased in a dose-dependent manner. CBZ concentration in teeth did not substantially change after death, with high CBZ distribution being observed in the pulp cavity. The tooth and serum CBZ concentrations were highly correlated, suggesting that the measurement of toxicant concentration in sampled teeth would allow for the estimation of blood toxicant concentration in highly decomposed corpses.

## 1. Introduction

Forensic examinations suspect the involvement of drugs and toxicants in all cases. There are many cases of drug-related deaths in the field of forensic medicine, such as drug-induced suicide [1], murder [2], medication errors [3,4], gas poisoning [5,6], and food poisoning [7,8]. The diagnosis of toxic effects requires the measurement of blood concentrations of the detected toxicants. In regular autopsies, blood and organs are used to quantify drug and toxicant concentrations; however, it is typically impossible to collect specimens from highly decomposed corpses. Furthermore, cadaver blood does not reflect the blood toxicant concentration at the time of death due to postmortem diffusion and redistribution [9]. Therefore, for the toxicological analysis of cadavers, samples that are highly preservable and resistant to postmortem effects are needed, which is why the use of hard tissue organs such as bones and teeth is attracting attention. Similar to hair and nails, the hard tissue in teeth accumulates drugs [10]. Teeth have a higher preservability than other organs and are expected to serve as alternative samples for drug quantification in highly decomposed cadavers, as deposited drugs or toxicants are likely to be protected from biological or chemical degradation [11,12]. In addition, teeth are the hardest structures in the human body and are not only resistant to decay but also to sample collapse due to external forces. However, there are only a few reports on the detection of drugs from teeth, such as the detection of morphine and cocaine from the teeth of drug abusers during dental treatment [13], nicotine and cotinine in deciduous teeth [14,15,16], and ethyl glucuronide (used to determine alcohol consumption behavior) from extracted teeth [17]. The presence of drugs of abuse in postmortem hard dental tissue has also been previously reported [18]. In these studies, the pulp containing blood vessels was not removed, and the mechanism through which the detected drug migrates to the tooth remains unclear. Nonetheless, a previous study has suggested that drugs are taken up during remineralization of teeth, as demonstrated by a remineralization reproduction model using pellets of solidified dentin powder from bovine teeth [19]. Furthermore, a case report [20] has revealed that drugs are most likely to be detected in the carious material of teeth collected from cadavers, followed by the root and crown. Currently, it is not possible to determine whether a drug is delivered from the blood circulation to the pulp cavity or if it is taken up during the remineralization process of teeth.

Moreover, to the best of our knowledge, no study has examined the correlation between tooth and blood drug concentrations, the process of drug distribution in teeth, and whether teeth could be useful for drug quantification. Hence, it is currently impossible to assess toxic effects from tooth samples. Therefore, herein, we examined the drug distribution in rat teeth to estimate the blood concentration of carbamazepine (CBZ) from the CBZ concentration in teeth. CBZ is a dibenzazepine with a molecular weight and formula of 236.27 and C_15_H_12_N_2_O, respectively. It is an effective antiepileptic drug and is also used to treat mania, the manic phase of manic-depressive psychosis, agitated states of schizophrenia, and trigeminal neuralgia and neuropathic pain [21]. Additionally, CBZ is the primary choice for bipolar disorder prophylaxis in the absence of therapeutic lithium response [22]. CBZ can be used by various patients, ranging from children to adults, and is widely used worldwide. We used CBZ in this study because its therapeutic and toxic ranges are close to each other, making it an important drug in forensic examinations.

## 2. Materials and Methods

### 2.1. Experimental Animals

Eight-week-old male Sprague Dawley (SD) rats were used as experimental animals. Their mandibular first molars and cardiac blood were used as tooth and blood samples, respectively. The teeth and blood samples were collected after euthanizing the animals. A tooth was gently extracted as follows: first, a weak wedge force was applied to the periodontal ligament space using an 18-gauge injection needle (Terumo Corporation, Tokyo, Japan) instead of a dental hevel to dislocate the tooth. Next, using a very small needle holder instead of dental forceps, the tooth was extracted by gently grasping the crown while being careful not to collapse the crown. At this time, it was confirmed that there was no breakage of the crown or root of the tooth. Teeth with broken crowns or roots were not used for any experiments, as broken fragments may have gone missing. Cardiac blood was collected as follows: First, the thorax of the animal was incised with a scalpel, and the skin, subcutaneous fat, and muscle were peeled off using a scalpel and forceps to expose it. The ribs were cut with forceps, the thoracic cavity was opened, an incision was made in the pericardium with a scalpel, and cardiac blood was collected using an 18-gauge needle (Terumo Corporation, Japan) and a 10 mL disposable syringe (Terumo Corporation, Tokyo, Japan) while observing the heart. This study was conducted following the approval and guidelines of our Institute’s Animal Experimentation Committee.

### 2.2. Drug Concentration Administered and Method of Administration

CBZ (Sigma-Aldrich, St. Louis, MO, USA) was suspended in physiological saline and administered once to the rats intraperitoneally using a disposable needle and syringe. Based on the human toxicity range and prior research [23,24,25], we set a CBZ administration concentration range between 20 and 500 mg/kg. Typically, the effective blood concentration range of CBZ is between 4 and 12 μg/mL, and the toxic range is over 12 μg/mL. Usual therapeutic doses of CBZ in humans range from 200 to 1200 mg/day, corresponding to approximately 3.3 to 20 mg/kg.

### 2.3. CBZ Extraction and Quantification from Teeth

The sampled teeth were washed three times with pure water. They were then cooled with liquid nitrogen and pulverized using a ball mill (Mixer Mill MM2, Retsch, Haan, Germany) for 15 min. CBZ-d10 (Sigma-Aldrich) was added as an internal standard before performing extraction, according to a previous report [19]. The pulverized teeth were suspended in 500 µL of methanol. Samples were vortexed and subjected to CBZ extraction in an ultrasonic bath for 60 min. During sonication, the temperature of water in the ultrasonic bath was maintained between 22 °C and 45 °C. After centrifugation of the samples, the supernatant was collected. To complete the extraction, the residues were re-suspended in 100 µL of methanol, and this process was repeated twice. Residues were vortexed and extracted in an ultrasonic bath (maintained at from 22 °C to 45 °C) for 60 min. Thereafter, the supernatants were combined. Supernatants collected from three extraction cycles were evaporated under a gentle stream of nitrogen at 40 °C. After evaporation, samples were dissolved in 50 µL of isopropanol. Liquid chromatography-mass spectrometry (LC-MS) (Waters Alliance 2695-ZQ, Waters, Milford, MA, USA) was used to determine the tooth and serum CBZ concentrations. To evaluate the accuracy and precision of the analytical method, control sample concentrations were measured daily for five consecutive days. Matrix effects, extraction recovery, and overall process efficiency were determined following the procedures outlined in a previous report [26].

### 2.4. Time Elapsed after CBZ Administration

To examine the impact of the difference in time elapsed after CBZ administration on its concentration in teeth and blood, rats were euthanized, and the tooth and blood samples were collected 15, 30, 60, 90, 120, and 180 min after administration of 100 mg/kg of the drug (*n* = 6 each group) (Figure 1).

### 2.5. Correlation between Tooth and Serum CBZ Concentrations

The Spearman correlation coefficient was used to determine the correlation between tooth and serum CBZ concentrations.

### 2.6. Drug Dose Administered

To examine the impact of administering different CBZ doses on tooth and blood concentrations, the rats were administered 20, 100, or 500 mg/kg of CBZ (*n* = 6 for each group). CBZ concentrations in the tooth and blood samples collected 60 min post-administration were measured (Figure 1).

### 2.7. Time Elapsed after Death

To examine the impact of time elapsed after death on CBZ concentration in teeth, rats were euthanized 60 min after administration of 100 mg/kg of CBZ, and the cadavers were kept at 22 °C for 0, 3, 7, and 15 days (*n* = 6 each group) (Figure 1). The tooth samples were then collected for CBZ concentration measurements. Cardiac blood samples were collected only on the day of euthanization (day 0), as further collection postmortem was not possible.

**Figure 1 diagnostics-13-00311-f001:**
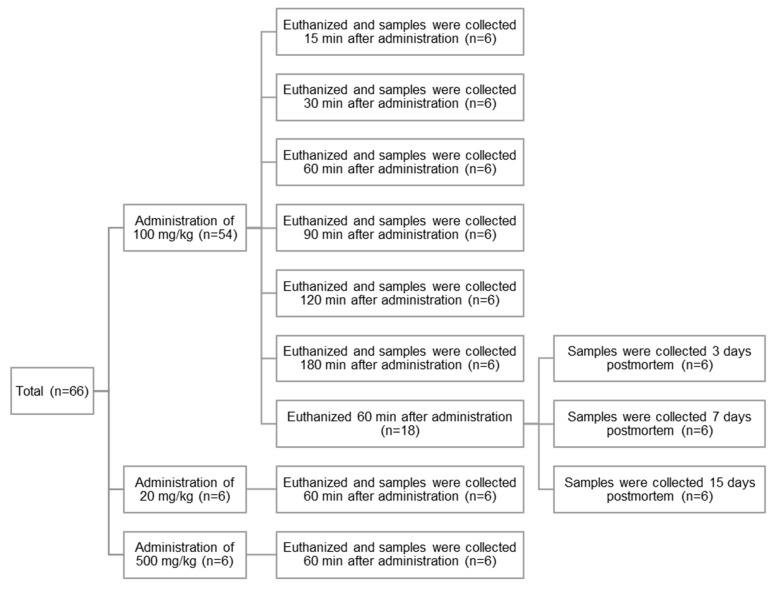
A schematic summary of rat samples.

### 2.8. Mass Spectrometry (MS) Imaging

Undecalcified, fresh, frozen sections were prepared from teeth sampled 60 min after administering 100 mg/kg CBZ to rats in accordance with the Kawamoto method [27,28,29,30]. Fresh frozen sections were made using adhesive film, with the following conditions: adhesive film (Cryofilm type 3C (16UF), 4D (16UF)) was used to observe the fine structure and weak fluorescence of the sample (the adhesive film was transparent and exhibited less autofluorescence). For MS imaging, a conductive adhesive film (Cryofilm type MS) was used to avoid sample charging. Biological samples were snap-frozen and cryo-embedded using a proprietary cryo-embedding medium (SCEM). Frozen samples were sliced with a fine and sharp disposable tungsten carbide blade (TC-65). Thick sections of 6 μm were made from rat tooth samples by using an adhesive film and a blade. Using an imaging mass microscope (iMScope, Shimadzu Corporation, Kyoto, Japan), the localization of CBZ was evaluated following the superimposition of CBZ distribution images obtained by optical and MS imaging. As a negative control, we prepared undecalcified, fresh, frozen sections of teeth sampled 60 min after administering physiological saline to rats.

### 2.9. Statistical Analysis

Data are presented as mean ± standard error (S.E.). Tukey’s multiple comparison test was used for the analyses of statistical significance. *p* values < 0.05 were considered statistically significant.

## 3. Results

### 3.1. Validation

Table 1 presents the validation of tooth CBZ concentration measurements using LC-MS.

### 3.2. Changes Due to Differences in Time Elapsed after Drug Administration

CBZ was detected in teeth 15 min after its administration, with its concentration peaking at 60 min post-administration and decreasing thereafter (Figure 2a). Serum CBZ concentration showed a similar trend in distribution, as observed with tooth CBZ concentration, and peaked at 60 min post-administration (Figure 2b).

Considering the possibility that CBZ may be detected in blood adhered to the tooth surface, we measured the CBZ concentration in the washing liquid remaining after the third tooth surface wash; we did not detect CBZ in the washing liquid.

### 3.3. Correlation between Tooth and Serum CBZ Concentrations

A comparison of tooth and serum CBZ concentrations showed a high correlation, as indicated by a correlation coefficient (r) of 0.8500 (Figure 3). The correlation formula was y = 1.4008x + 4.0726.

### 3.4. Changes Due to Differences in Dose Administered

CBZ was detected in the teeth of rats administered 20 mg/kg of the drug. Tooth CBZ concentration increased in a manner dependent on the concentration of CBZ administered (Figure 4a). Serum CBZ was also detected in rats administered 20 mg/kg of the drug, showing a dose-dependent relationship (Figure 4b).

### 3.5. Postmortem Changes in Tooth and Serum CBZ Concentrations

Rats euthanized 60 min after the administration of 100 mg/kg of CBZ did not show significant differences in tooth CBZ levels on days 0, 3, 7, and 15 postmortem (Figure 5).

### 3.6. Localization of CBZ in Teeth

The MS images of the teeth of rats euthanized 60 min after receiving CBZ (100 mg/kg) showed a high distribution of the drug in the pulp cavity (Figure 6).

## 4. Discussion

Our study findings revealed for the first time that systemic CBZ distributes to the teeth. In addition, this is the first report that demonstrates the migration of drugs to the teeth.

The distribution of CBZ to the teeth had similar dynamics to its distribution to cardiac blood, suggesting that distribution of drugs to the teeth may be a consequence of its migration from the blood. Although a report has suggested that drug distribution to teeth occurs during remineralization [19], the results of this study confirmed the localization of CBZ in the pulp cavity, a region that houses the vasculature of teeth. In addition to the correlation between tooth and serum CBZ concentrations, our findings support the theory that a drug’s distribution to the teeth can be attributed to its migration to the pulp cavity via blood vessels.

Visualizing the tissue localization of target compounds using MS imaging began to attract attention in 2001 when this technique was used by Stoeckli et al. [31]. Its application has since spread rapidly, with improvements in the instruments used. In the area of forensic medicine, using MS imaging for the analysis of drug localization in hair samples to estimate the timing of drug intake by drug abusers has been reported [32]; however, our study is the first to demonstrate that MS imaging can be used to analyze drug localization in teeth.

In addition, because the CBZ concentrations in teeth and cardiac blood showed a high correlation and the CBZ concentration in teeth was stable even after death, our study demonstrates that it is possible to estimate blood drug and toxicant concentrations of highly decomposed cadavers using teeth. We reported the following correlation formula: y = 1.4008x + 4.0726; using this formula, the blood concentration (y) can be estimated by measuring the concentration of toxicant in the extracted tooth (x). By conducting similar research on other drugs, we aim to establish a comprehensive method for estimating blood drug and toxicant concentrations from teeth. Additionally, by conducting similar experiments with bones, which are also hard tissues, it is anticipated that the accuracy of estimations of blood drug concentration from highly decomposed carcasses such as skeletons can be further improved. In the human body, bones also have high preservability. As such, a recent case study reported the detection of CBZ in human ribs [33]. However, since it is necessary to consider the problems related to dosage, the difference in elapsed time after administration, the method of administration, the presence or absence of individual diseases, the circumstances leading to death, and the accuracy of the information, a huge number of samples are required to estimate blood drug concentrations from tooth samples. In animal experiments, these conditions can be matched. In the future, we will conduct similar experiments using rat bones, and by combining the results of this experiment, we believe that we can obtain a more accurate correlation formula for the determination of blood drug concentrations between the investigated organs. In this study, experiments were only conducted at 22 °C. In the future, evaluation in various environments is necessary to determine the effects on drug recovery. We also anticipate that it will be possible to estimate blood drug concentrations more precisely. Furthermore, we will verify whether this correlation formula can be extrapolated for application in humans. Moreover, depending on the drug being tested, our study may help in the development of new treatment methods in clinical practice. For example, the “standard of care” for irreversible pulpitis is the immediate removal of the pulp from an affected tooth and is widely accepted; however, in certain parts of the world, antibiotics continue to be prescribed [34]. A novel effective antibiotic treatment strategy against pulpitis can be developed if it is proven that a sufficient level of the antibiotic reaches the dental pulp. 

## 5. Conclusions

CBZ concentration in the teeth and serum of rats administered the drug showed similar dynamics and localization, with CBZ levels remaining stable in the teeth even after death. However, only one type of drug was used and monitored up to 15 days after death in this study. Further investigations with long-term experimental periods and other drugs are necessary to ensure reliable and robust blood drug concentration estimation using teeth. In the future, our findings depicting that drugs migrate to the dental pulp can be applied to research using antibacterial drugs. This research can be applied not only to the field of forensics but also in the clinical field. Our findings may help in the development of new and standardized treatment strategies for pulpitis.

## Figures and Tables

**Figure 2 diagnostics-13-00311-f002:**
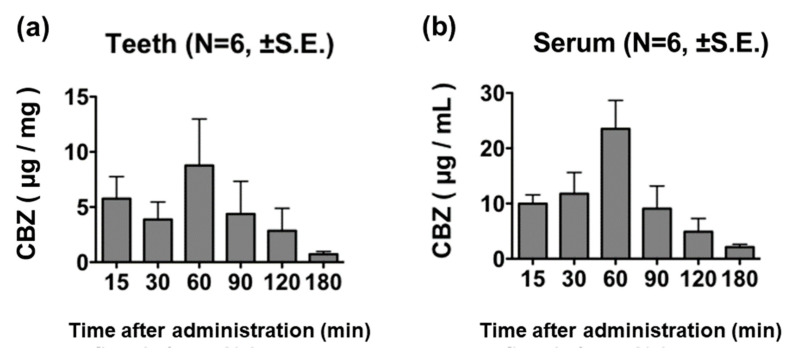
CBZ concentration in teeth and cardiac blood samples after the administration of 100 mg/kg of CBZ. The CBZ concentration in teeth (**a**) and cardiac blood (**b**) showed a similar trend, both peaking at 60 min post-administration.

**Figure 3 diagnostics-13-00311-f003:**
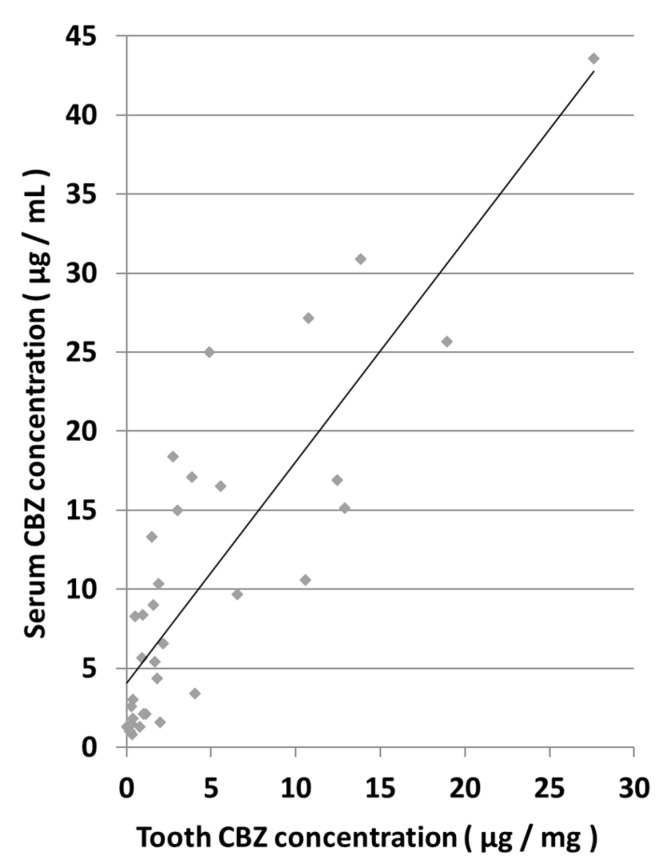
Correlation between tooth and serum CBZ concentrations.

**Figure 4 diagnostics-13-00311-f004:**
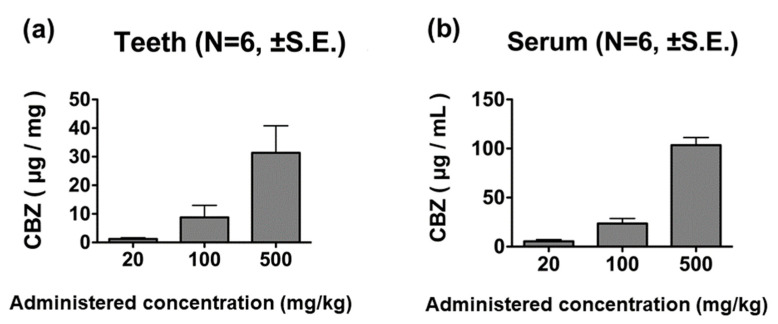
CBZ concentrations in tooth and blood samples 60 min after administration. The CBZ concentration in teeth (**a**) and cardiac blood (**b**) increased in a dose-dependent manner.

**Figure 5 diagnostics-13-00311-f005:**
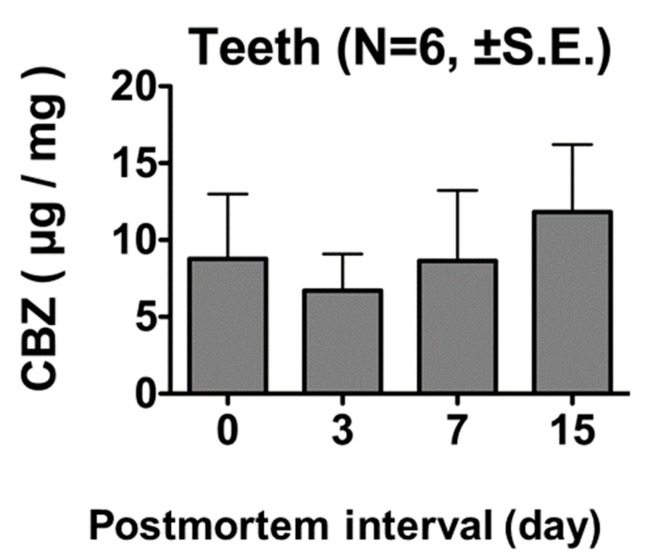
CBZ concentrations in tooth samples 60 min after administration of 100 mg/kg CBZ. No significant differences were observed in tooth CBZ concentration on days 0, 3, 7, and 15 postmortem.

**Figure 6 diagnostics-13-00311-f006:**
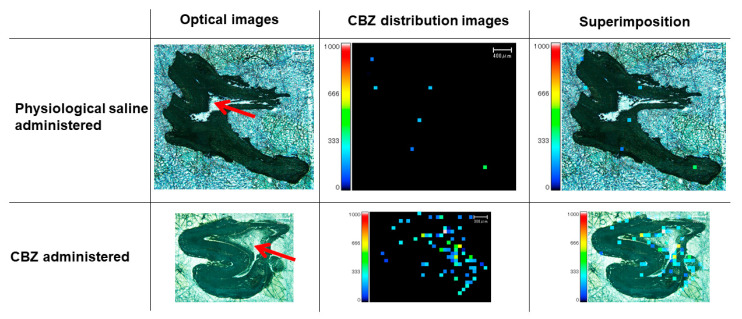
MS imaging of teeth 60 min after administration of 100 mg/kg CBZ. High distribution of CBZ in the pulp cavity is depicted. Red arrow, pulp cavity. Red and yellow squares, high concentration of CBZ.

**Table 1 diagnostics-13-00311-t001:** Validation of LC-MS results.

Validation Parameter	Values
Limit of detection	0.025 μg/mg
Limit of quantitation	0.1 μg/mg
Precision (RSD) (at 0.1 µg/mL, interday, *n* = 5)	18.40%
Accuracy (bias) (at 0.1 µg/mL, interday, *n* = 5)	14.44%
Matrix effect (*n* = 5)	95.95 ± 2.76 (%)
Recovery (*n* = 5)	59.13 ± 3.41 (%)
Process efficiency (*n* = 5)	56.74 ± 3.28 (%)

## Data Availability

Not applicable.

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
