# Peer review of "Application of Teeth in Toxicological Analysis of Decomposed Cadavers Using a Carbamazepine-Administered Rat Model"

_diagnostics, 2023, doi:10.3390/diagnostics13020311_

Round 1
Reviewer 1 Report
Comments
1. Grammatical mistakes should be corrected
2. References should be as per journal guidelines
3. Novelty of the work is not mentioned.
4. Not a very unique study. These type of forensic study are already reported.
Author Response
Reviewer #1:
We are grateful for the reviewer’s critical comments and useful suggestions that have helped us improve our manuscript. As indicated in the following responses, we have considered the comments and suggestions and have revised our manuscript accordingly.
Comments:
- Grammatical mistakes should be corrected
- References should be as per journal guidelines
- Novelty of the work is not mentioned.
- Not a very unique study. These type of forensic study are already reported.
Responses to comments:
1. We have used an English editing service to re-edit the manuscript.
2. We have edited the references per the journal guidelines.
3 and 4. This research used teeth as the subject. It is therefore novel in comparison with previous forensic research in that the findings have potential application not only in the field of forensic medicine but also in dental clinical practice. To this end, we have added the following sentence in the Discussion:
“For example, the “standard of care” for irreversible pulpitis is immediate removal of pulp from an affected tooth and is widely accepted; however, in certain parts of the world, antibiotics continue to be prescribed [34]. A novel effective antibiotic treatment strategy against pulpitis can be developed if it is proven that a sufficient level of the antibiotic reaches the dental pulp.”
In addition, we changed the title of the manuscript to read as follows:
Application of teeth in toxicological analysis of decomposed cadavers using a carbamazepine-administered rat model
Reviewer 2 Report
Comments are in the pdf

Author Response
Reviewer #2:
We are grateful for the reviewer’s critical comments and useful suggestions that have helped us improve our manuscript. As indicated in the PDF, we have considered the comments and suggestions and have revised our manuscript accordingly.
Comments and responses:
- Comment: language editing is required
Response: We have used an English editing service to re-edit the manuscript.
- Comment: Keyword must be in alphabetical order
Response: We have arranged the keywords in alphabetical order.
- In line 45, “there are few reports” has been changed to “there are only a few reports”
- Comment: In Line 56, sentence is not clear
Response: We modified the sentence to read as follows: “Furthermore, a case report [20] has revealed that drugs are most likely to be detected in the carious material, followed by the root and crown, of teeth collected from cadavers.”
- In line 64, “dibenzazepine” was spell checked.
- In line 67, we added a reference.
- Comment: total sample size and how the groups were divided must be elaborated in detail. Provide a schematic summary of the samples
Response: We have included Figure 1 that shows a schematic summary of rat samples
- Comment: In Line 99, “WHY SUCH A WIDE RANGE? Each rat was given how much?”
Response: Because we aimed to examine the impact of the difference of CBZ doses on tooth and blood concentration, the rats were administered 20, 100, or 500 mg/kg of CBZ. This is further described in Section 2.6. “Drug Dose Administered.”
- Comment: In Line 123, “This was also after euthanizing right? Below u have mentioned that 60 min after drug administration the animals were euthanized, so just wanted to confirm about the 15, 30, 90, 120, 180 min”
Response: We have accordingly revised this part as follows in Line 121: “rats were euthanized, and the tooth and blood samples were collected 15, 30, 60, 90, 120, and 180 min after administration”
- Comment: In Line 139, “in reality as the body decomposes the tooth is not going to be kept separately in a 22 degree C right? So how will this study results translate in a real life scenario? Justify preserving the tooth at 22 degree celcius.”
Responses: We kept the cadavers with their teeth at 22 °C. Teeth were not separated from the body. We have changed “To examine the impact of the difference in time elapsed after death on the tooth CBZ concentration, rats were euthanized 60 min after administration of 100 mg / kg of CBZ and kept at 22 °C for 0, 3, 7, and 15 days (n = 6 each group)” to “To examine the impact of time elapsed after death on CBZ concentration in teeth, rats were euthanized 60 min after administration of 100 mg/kg of CBZ, and the cadavers were kept at 22 °C for 0, 3, 7, and 15 days (n = 6 each group) (Fig. 1).” In addition, as the reviewer pointed out, in a real life scenario, there is no environment that can be maintained at 22 °C all the time. However, in this study, the air temperature had to be controlled to ensure experimental reproducibility. In future, we believe that evaluations in various environments will be necessary. We have therefore added the following sentence in the Discussion: “In this study, experiments were only conducted at 22 °C. In future, evaluation in various environments is necessary to determine the effects on drug recovery.”
- In line 220, “apparatuses” was changed to “instruments.”
- In line 223, “is” has been retained, as suggested by the language editing service.
- In line 235, “white bone” was changed to “”
- In line 235, in response to the advice from the reviewer, “bones have the second highest preservability next to teeth” was changed to “bones have also high preservability.”
- In line 250, in response to the advice from the reviewer, we added the reference and following sentence in the Discussion. “For example, the “standard of care” for irreversible pulpitis is immediate removal of pulp from an affected tooth and is widely accepted; however, in certain parts of the world, antibiotics continue to be prescribed [34]. A novel effective antibiotic treatment strategy against pulpitis can be developed if it is proven that a sufficient level of the antibiotic reaches the dental pulp.”
- In line 258, in response to the advice from the reviewer, we added the limitation of our study as follows in the Conclusion. “However, only one type of drug was used and monitored up to 15 days after death in this study. Further investigation with long-term experimental periods and other drugs is necessary to ensure reliable and robust blood drug concentration estimation using teeth.”
Round 2
Reviewer 2 Report
Revisions are to my satisfation